# The UAS-Based 3D Image Characterization of Mozarabic Church Ruins in Bobastro (Malaga), Spain

**Carlos Enríquez** [1,*], **Juan Manuel Jurado** [2], **Alexandro Bailey** [3], **Danilo Callén** [3], **María José Collado** [4], **Gabriel Espina** [3], **Pablo Marroquín** [3], **Erick Oliva** [3], **Edgar Osla** [3], **María Isabel Ramos** [1], **Scarlett Sarceño** [3] and **Francisco Ramón Feito** [2]

[1] Department of Cartographic and Geodetic Engineering and Photogrammetry, University of Jaén, 23071 Jaén, Spain; miramos@ujaen.es

[2] Computer Graphics and Geomatics Group of Jaén, University of Jaén, 23071 Jaén, Spain; jjurado@ujaen.es (J.M.J.); ffeito@ujaen.es (F.R.F.)

[3] Facultad de Arquitectura, Universidad de San Carlos de Guatemala, Ciudad de Guatemala 01012, Guatemala; 201016663@farusac.edu.gt (A.B.); danilo.callen@farusac.edu.gt (D.C.); 201315093@farusac.edu.gt (G.E.); 201213879@farusac.edu.gt (P.M.); 201214188@farusac.edu.gt (E.O.); 201314979@farusac.edu.gt (E.O.); 201410290@farusac.edu.gt (S.S.)

[4] Department of Historical Heritage, University of Jaén, 23071 Jaén, Spain; mcollado@ujaen.es

\* Correspondence: cenrique@ujaen.es

**Abstract:** In recent years, the application of geomatics tools in archaeology has proved to be very useful to obtain meaningful knowledge of the 3D reconstruction of archaeological remains and semantic classification of the 3D surface. These techniques have proven to be an effective solution for the 3D modeling and the extraction of many spatial features on an archaeological site. However, novel methodologies as well as new data exploitation strategies are required to exploit these geospatial data for natural and cultural heritage documentation, monitoring, and preservation. In this paper, we have studied unique archaeological ruins, a Mozarab church in Al-Andalus, using high-resolution RGB images, which was taken by a drone. Thus, a 3D reconstruction of the ruins and the surrounding environment is carried out in order to characterize it on a dense point cloud. Then, a digital elevation model (DEM) was calculated in order to identify critical slope lines, which are significant to determine where the structure of the church was built. Our results can be used for the development of an architectural project and thus a virtual recreation of these archaeological ruins was performed.

**Keywords:** remote sensing; archaeology; geomatics; 3D reconstruction; UAS

## 1. Introduction

At the end of the 9th century, Umar ibn Hafsun rose up against Cordoba's emirate and set up its capital in the ruins of an old Roman castle in Bobastro, near Ardales (Malaga). Umar ibn Hafsun converted to Christianity around 899, gaining the support of the Mozarabs, which are Iberian Christians who lived under the Moorish rule in Al-Andalus, Malaga's population and maybe with the purpose of getting help from Alphonse III. However, this decision was the beginning of his end, since most of his followers, Berbers, and Muladis who stayed faithful to Islam turned their backs on him. In 912, Abd al-Raḥmān III became emir and decided to pacify al-Andalus. He attacked Bobastro in an expedition where he was able to conquer more than 70 fortresses. The attacks were ongoing up to 917 when Umar ibn Hafsun passed away. Finally, in 928, his son Suleyman was defeated in Bobastro by Abd al-Raḥmān III who destroyed the whole city after taking total control of the enclave. Nowadays, the only remains of this uprising are the ruins of a Moorish fortress, a Muslim necropolis, and the only Mozarab church in Al-Andalus.

Nowadays, to apply Unmanned Aerial Systems (UASs) to archaeology is no longer a novelty because its use in archaeological fieldwork has been increasing exponentially in the last decade. A full description of state-of-the-art drones in archaeology can be found in [1]. The use of data from remote sensing techniques such as aerial photography and satellite imagery, as well as Digital Elevation Models (DEM), are already firmly established as techniques for studying archaeological landscapes [2]. In addition, UAS can be used for the creation of orthophotos and DEMs as well as the identification of archaeological features and microrelief, as proxy indicators of the presence of archaeological buried remains [3,4]. The accuracy of DEMs obtained by UAS is studied in [5] just as the influence of the number of Ground Control Points (GCPs) used for georeferencing on Digital Surface Model (DSM) and orthoimage accuracies obtained by UAS-photogrammetry is studied in [6]. How the SfM technique allows reconstructing landform topography based on UAS images can be seen in [7].

The use of photogrammetry is increasingly applied for the preservation of cultural heritage. This technique is a common practice for the acquisition of real-world assets with a highly realistic appearance. The use of high-resolution cameras enables through photogrammetry to scan a high degree of fidelity of the scene. In addition, drones play a key role for the 3D reconstruction in archaeology. Frequently, the archaeological remains are in places with a difficult access and irregular terrain, so UAV-based solutions are a better option than LIDAR [4,8]. UAVs can capture overlapping images from multiple viewpoints surrounding the target object. In this study, we have applied the structure-from-motion (SfM) method to create a 3D model of the archaeological ruin. SfM photogrammetry employs an automated process to identify and match key points between overlapping images [9]. Thus, a 3D point cloud was generated in order to study the geometric characterization of the studied archaeological ruins.

This paper deals with 3D reconstruction and visualization of the heritage site of Bobastro and how can it be useful with identification of the main elements of the Bobastro Mozarab church. Our goal was to expand on the thorough contributions made by the first archaeologist [10–12], who studied this area at the beginning of the 20th century, with the help of this new methodology that facilitates the analysis and approach of interpretive hypotheses. Thus, our purpose was to restore the lost images of what for almost a century had been abandoned, robbed and mistreated ruins.

The generation of the DEM allows us to identify the main axes of the buildings and the slopes. The analysis of the slopes will lead us to define the distribution of the outer spaces and the circulation between the temple and the cloister.

The main contributions of the proposed methodology are: (1) the identification of the axis, (2) the identification of the cloister, and (3) the identification of minor structures, such as walls, a cistern, a grave, etc. This article is structured as follows: Section 2 describes the study area and the sensor used for data collection. Section 3 shows the proposed methods for data acquisition, 3D reconstruction, point cloud characterization, orthophoto, and DEM generation. In Section 4, the results are presented and main issues are discussed in Section 5. Finally, Section 6 presents the main conclusions and further research.

## 2. Materials

### 2.1. Study Area

The archaeological site of Bobastro is located in an area known as Mesas de Villaverde, in the northwest of the province of Malaga, on the southern Mediterranean coast of Spain. Its geographic coordinates are 36°54′N and 4°46′ WG, referred to ETRS89, and about 500 m above sea level. The vegetation consists mainly of the Mediterranean pine. The topography in the study area is mountainous with steep slopes. In fact, its sheer orography was paramount when the rebels chose it as a natural stronghold. The site is disposed directly on a subhorizontal tortonian unit [13]. Lithologically, it is a thick post-orogenic unit made up of conglomerates and calcarenites, lying unconformably over

internal zones of the Betic Cordilleras. For the purpose of our research, we covered an area of 0.16 ha. Figure 1 shows the study area in which this research was conducted.

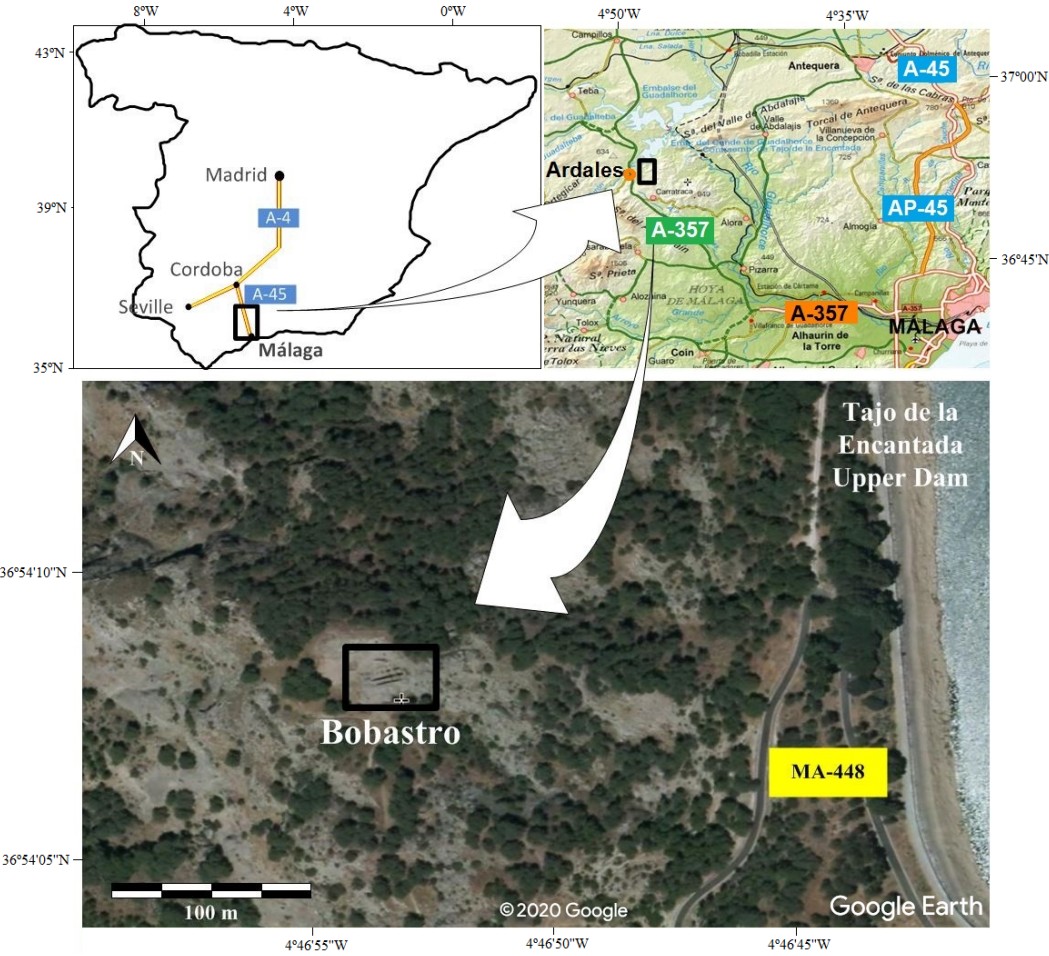

**Figure 1.** Clockwise from top left. Research area. The upper images represent the location of our area of research in Spain. The lower image, taken from Google Earth, is the precise localization of Bobastro. The coordinates (m) are UTM, zone 30, referred to ETRS89.

## 2.2. UAS-Based and Field Data

An Unmanned aircraft system (UAS) is, according with the definition given by the International Civil Aviation Organization (ICAO), an aircraft and its associated elements operated with no pilot on board [14]. Popularly known as drones, RPAs (remotely-piloted aircraft systems), or UAVs (unmanned aerial vehicles), are the latest non-invasive methods used for archaeological research. The use of UASs is extremely functional, efficient, and cheap [3], and in the field of cultural heritage its purpose ranging from the documentation to the detection of archaeological features. Among its advantages are: the capacity to carry a great variety of sensors (thermal, RGB, or LIDAR among others), they are user-friendly, and have a high quality of the obtained products. According to the ICAO guidelines [14], we will reserve the use of UAVs just for the aircraft, while the term UAS will refer to the whole: aircraft and its associated elements. In this research, the aerial survey was undertaken with a DJI Phantom 4 Pro, equipped with a DJI FC6310 camera with an 8.8 mm nominal focal length, and a 1″ CMOS 20 megapixel sensor with 2.41 × 2.41 μm nominal pixel size. The DJI FC6310 camera creates an image of 5472 × 3648 pixels corresponding to 13.2 × 8.8 mm [15]. The camera parameters are described in Table 1.

**Table 1.** Main characteristics of the camera.

| Camera Characteristics | |
| --- | --- |
| Camera | DJI FC6310 |
| Sensor | 1″ CMOS 20 megapixel |
| Nominal focal length | 8.8 mm |
| Pixel resolution | 5472 × 3648 pixels (13.2 × 8.8 mm) |

To get the coordinates of the GCPs, we used a single Topcon GR5 RTK-GNSS, with centimeter accuracy linked to the Andalusian Positioning Network (Red Andaluza de Posicionamiento, RAP), an active geodetic reference frame materialized by 22 permanent reference stations. The RAP-NRTK solution provides a precision between 0.004 m and 0.030 m in the East component, between 0.004 m and 0.059 m in the North component and between 0.007 m and 0.094 m in the Up component [16]. Therefore, we were able to obtain coordinates with a centimeter accuracy which are acceptable, since the range of the uncertainty of the acquisition and measurement of the objects (e.g., vegetation, stones, loose soil) with the photogrammetric survey is greater than a centimeter [4].

## 3. Methods

### 3.1. Getting Data

Following the sequence described in [17], we began getting coordinates of the GCPs. This is an important aspect of the data collection process because it enables an accurate geolocation and geometric quality. Therefore, it was necessary to design the distribution of these points on the ground, with known coordinates, in order to get the results georeferenced with a national grid and a height datum. Unless otherwise indicated, all coordinates are in meters, the map projection is the UTM, the references system is ETRS89 and the heights are referred to the mean sea level in Alicante (Spain). Since the area was rather small, about 2500 m$^2$, and, taking into account our own experience, we decided to take six GCPs, from which four were used to georeference the images and the other two, the inner ones, were used as control points. Once the survey was done, we began with the flight. It was carried out in manual mode at altitudes around 100 m above sea level, 70% longitudinal, and 80% lateral overlap of images.

All the images were acquired by adjusting the optical axis in vertical position. The final ground sampling distance (GSD) was 2.7 cm/px. The photogrammetric results are highly influenced by the quality of input images. Therefore, sensors, settings, and acquisition plans had to be considered to ensure optimal image data. The weather was partially cloudy and the flight was conducted close to the solar noon time in order to minimize shadows and specular lighting. On one hand, there were no shadows that could mask details and, on the other hand, we were forced to use an ISO 500. The aperture was set to f/6.3 and exposition time was 1/500 s. The time flight was about 25 min during with 257 images were taken. The trajectory of the UAS and the position of the GCPs can be seen in Figure 2.

Once the data were taken, the RGB images were used to generate a dense point cloud. Next, with the GCP coordinates, the point cloud was referenced, and finally the DEM and the orthophoto were created. Once all the information was gathered, and with the help of the field notes, the analysis and interpretation processes could be carried out. The general flow of the whole process can be seen in Figure 3.

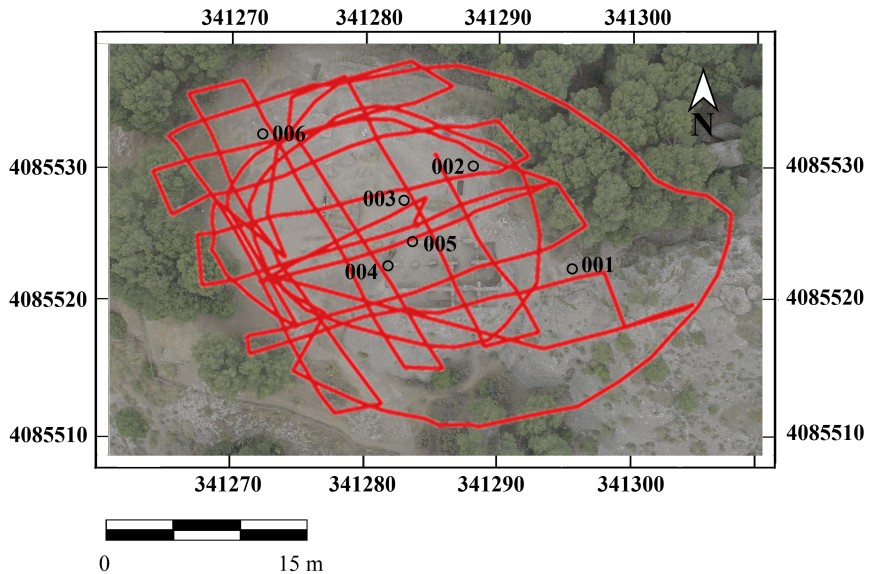

**Figure 2.** Trajectory of the UAS (red) and position (black) of the GCPs in the study area. The coordinates (m) are UTM, zone 30, referred to as ETRS89.

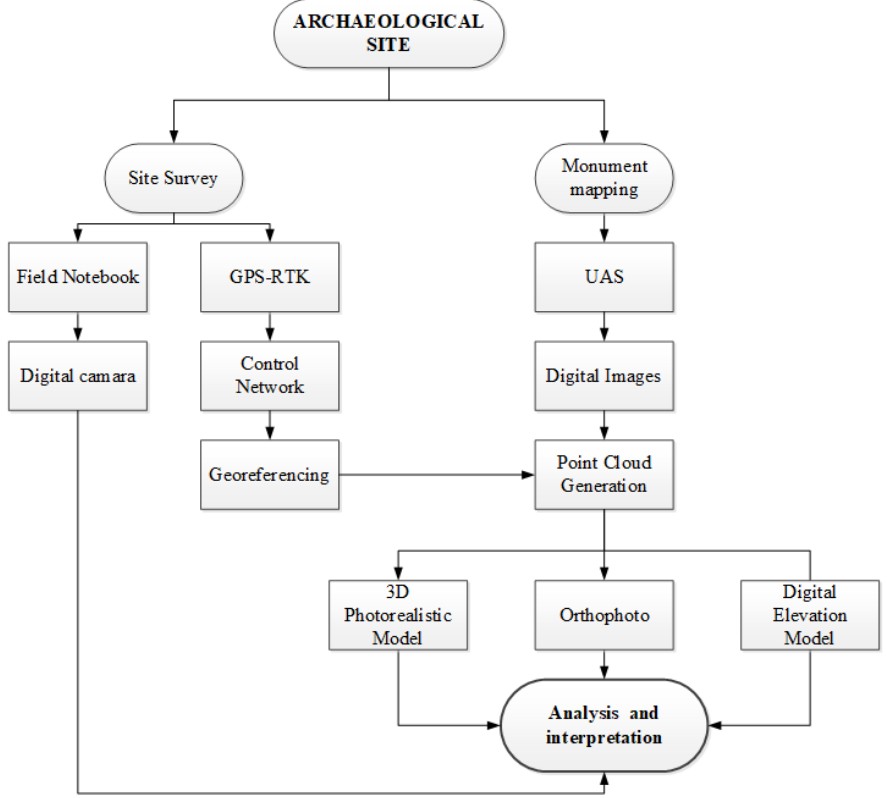

**Figure 3.** The general workflow of the proposed methodology.

*3.2. Ground Control Point: GNSS Measurements*

The use of measurements with the GNSS receiver to give exact coordinates of the points used for the correct geolocation of the images captured by the UAS has already been explained. Furthermore, these points allow us to analyze the geometric quality of the model in terms of preserving the scale.

The quality of the 3D model scale was validated using the root-mean-square error (RMSE), which was calculated for the X and Y axes (Equation (1)) and the Z axis (Equation (2)).

$$RMSE_{X,Y} = \sqrt{\frac{1}{n}\Sigma_{i=1}^{n}\left(X_i - X_{i,GNSS}\right)^2 + \left(Y_i - Y_{i,GNSS}\right)^2}$$

(1)

$$RMSE_Z = \sqrt{\frac{1}{n}\Sigma_{i=1}^{n}\left(Z_i - Z_{i,GNSS}\right)^2}$$

(2)

where: $X_{i,ref}$, $Y_{i,ref}$ and $Z_{i,ref}$ are the coordinates of the 3D point measured by the GNSS and $X_i$, $Y_i$ or $Z_i$ are the coordinates of point *i* on the point cloud. Table 2 shows the coordinates of the GCPS and the errors associated which each point, and Table 3 shows the RMS error associated with the control points and the check points.

**Table 2.** UTM Coordinates (m) and associated errors ($\delta$ E, $\delta$ N, $\delta$ H) of the GCPs. Errors are measured in centimeters.

| Label | E (m) | N (m) | H (m) | $\delta$ E | $\delta$ N | $\delta$ H | Total $\delta$(cm) |
|---|---|---|---|---|---|---|---|
| **Control points** | | | | | | | |
| 001 | 341,309.363 | 4,085,524.571 | 504.181 | −1.4 | < 0.1 | 1.1 | 1.8 |
| 002 | 341,296.562 | 4,085,534.953 | 504.180 | −1.6 | 2.2 | 1.3 | 3.0 |
| 004 | 341,287.024 | 4,085,519.773 | 503.365 | 3.5 | 1.9 | 0.2 | 4.0 |
| 006 | 341,273.320 | 4,085,535.914 | 503.336 | −1.7 | 1.3 | 2.3 | 3.1 |
| **Total** | | | | **2.2** | **1.6** | **1.4** | **3.1** |
| **Check points** | | | | | | | |
| 003 | 341,290.016 | 4,085,531.161 | 502.767 | 5.2 | −2.4 | −5.7 | 8.1 |
| 005 | 341,291.095 | 4,085,524.135 | 504.225 | −4.0 | −3.0 | 0.7 | 5.0 |
| **Total** | | | | **4.6** | **2.7** | **4.1** | **6.7** |

**Table 3.** RMS errors (cm) in each component associated with the GPCs.

| Count | E | N | H | EN | Total |
|---|---|---|---|---|---|
| **Control points** | | | | | |
| 4 | 2.2 | 1.6 | 1.4 | 2.7 | 3.1 |
| **Check points** | | | | | |
| 2 | 4.6 | 2.7 | 4.1 | 5.4 | 6.8 |

### 3.3. Point Cloud Reconstruction

In this paper, the photogrammetric processing is applied using the SfM method [18]. It is an open-source method, which can detect the same regions of overlapping images, determine their geometric relationships, and infer the rigid scene structure (point set) with the pose (position and orientation) of all cameras. The resulting 3D structures are modeled as a point cloud instead of a 3D mesh due to the fact that many complex geometric objects cannot be correctly triangulated. The SfM has been used for multiple purposes such as the vegetation reconstruction [19], cultural heritage [20], and archaeology [21]. Focusing on this last one, the 3D modeling of archaeological artifacts is usually carried out to study different characteristics using a virtual copy instead of the original. The use of UASs and photogrammetry techniques in archaeology implies a full transformation for the acquisition of archaeological knowledge.

To apply SfM photogrammetry in archaeology, some relevant aspects to obtain an accurate survey are as follows: (1) any feature to be reconstructed should be visible in at least three images, (2) the study area should be sufficiently illuminated, and (3) the image scale should be 1/2 or 1/4 to recognize more key points.

The processing of the images was performed using the specific software Agisoft Metashape (v. 1.5.5.9097). This software allows the reconstruction of a 3D model from the images by applying the SfM algorithms. The alignment of the images and the 3D reconstruction are fully automated, and the images can be taken in any position, the only condition being that at least two images are necessary for the same point. It also can be used to generate a dense point cloud and make geometric corrections in the resulting 3D model. Processing of images with Metashape includes the following main steps (see Figure 4):

- Loading images into Metashape: Before starting any operation, it is necessary to point out which images will be used as a source for photogrammetric processing. After inspecting loaded images, the unnecessary ones are removed.
- Input coordinate data: To reference the model, coordinates of at least three points on the scene should be specified.
- Identify GCPs in the images: This step will be useful if the alignment process fails.
- Aligning images: Once photos are loaded, Metashape needs to find common elements in two different images. This process is done automatically; however, if it fails you will have to do it manually, setting markers (at least 4 per photo) in these photos and indicating their projections on at least two photos from the already aligned subset. At that point, it is always a good idea to use GCPs as a markers wherever it is possible.
- Remove noise: To remove undesired data, like people or vegetation, the data must be cleaned and filtered. In addition, spurious data can be removed after the point cloud registration, if they may prove useful for registration in the absence of good overlaps, or if there is little ground control. To do so, first non-static objects during the data acquisition process should be removed from the images. Secondly, after the alignment is done and the sparse point cloud creates mislocated points, they should be deleted. As a tourist area, the ruins are visited by many people, which means there were visitors going back and forth during the survey. Therefore, we had to remove a large number of 'ghost images' in the images, before the point cloud generation. Nevertheless, the generated point cloud had a large amount of noise caused by the traffic of the people who were wandering about. It was necessary to perform an automatic cleaning of the point cloud by statistical filters to eliminate 'out layers'.
- Optimizing camera alignment: Possible nonlinear deformations of the model can be removed by optimizing the estimated point cloud and camera parameters based on the known reference coordinates. During this optimization, Metashape adjusts estimated point coordinates and camera parameters minimizing the sum of reprojection errors and reference coordinate misalignment errors [22].
- Building a dense point cloud: Based on the estimated camera positions, Metashape calculates depth information for each camera to be combined into a single dense point cloud.
- Building mesh (3D polygonal model): After a dense point cloud has been reconstructed, it is possible to generate a polygonal mesh model based on the dense cloud data.
- Building digital elevation model (DEM): A georeferenced DEM is generated from the dense point cloud. It will serve as a basis for the orthopmosaic.
- Building orthomosaic: The orthomosaic is obtained by orthorectification of the original images, providing the scale is uniform throughout the image and, therefore, can be used to measure real distances.

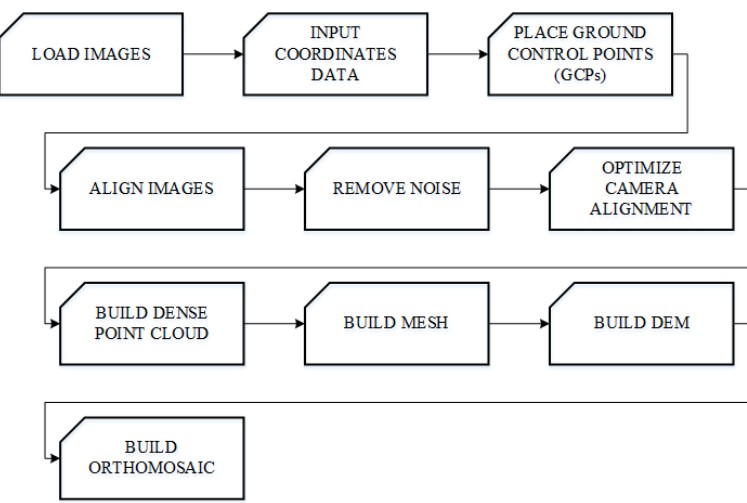

**Figure 4.** Workflow of the point cloud reconstruction process.

### 3.4. The Digital Elevation Model (DEM) and the Orthomosaic

From the dense point cloud, we created a mesh to give it texture in order to obtain a representation of the reality as faithfully as possible. As our final idea was to use these results in a Virtual Reality environment, we were forced to generate the mesh with ultra high quality. Therefore, the data became too large and the process was highly time-consuming, but at least the final result was worth it (see Figure 5).

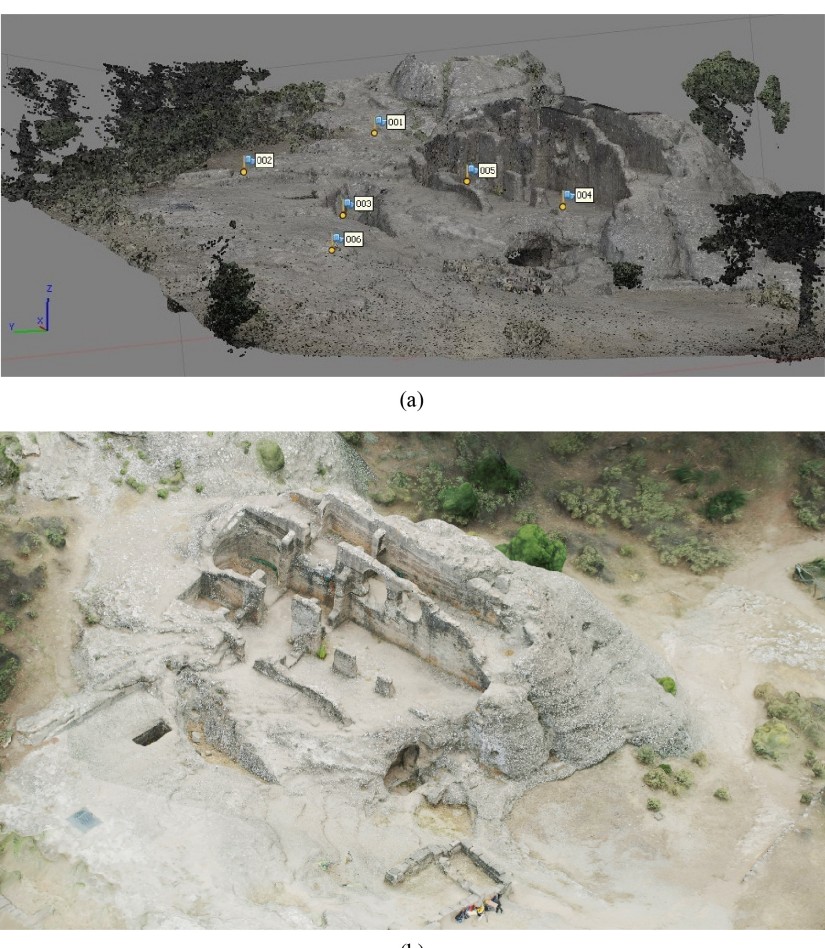

**Figure 5.** The 3D model: (**a**) the dense point cloud and (**b**) the reconstructed mesh in an isometric view.

The first step to create a DEM was to classify all the points in the dense cloud points in order to isolate the ground points from the rest of the points like vegetation or artificial structures because failing to do so results in distortions appearing during the DEM and the orthomosaic generation process. An orthomosaic is just a raster image created by merging orthophotos.

Once the DEM was created, we finished this method by creating an orthomosaic. The orthomosaic generation is based on orthorectification, which is the process for the removal of effects in an image perspective (tilt) and relief (terrain). Thus, a planimetrically correct image can be obtained. The resulting orthorectified image has a constant scale, wherein features are represented in their 'true' position [23].

## 4. Results

The 3D reconstruction process requires high hardware specifications in order to get a reasonable processing time. In Table 4, there are the main characteristics of the computer that we used for the data processing. It must be pointed out that the generation of the dense point cloud process is very time consuming.

**Table 4.** A summary of computer specifications.

| Hardware | Specification |
| --- | --- |
| RAM | 31.64 GB |
| CPU | Intel(R) Core(TM) i7-8700 CPU @ 3.20 GHz |
| GPU(s) | GeForce GTX 1060 6 GB |

Table 5 is a summary of the flying and data processing parameters and the time needed for each process. Looking at the results, it seems reasonable that, in the first trials, the models should be generated in low quality.

**Table 5.** Profiling of each stage.

| **Flight Parameters** | |
| --- | --- |
| Flight time | 24 min |
| Flying altitude | 100 m |
| Number of images | 279 |
| **Dense Point Cloud Generation** | |
| Points | 111,045,063 |
| Point colors | 3 bands, uint 8 |
| **Depth maps generation parameters** | |
| Quality | Ultra High |
| Filtering mode | Moderate |
| Processing time | 16 h 13 min |
| **Dense cloud generation parameters** | |
| Processing time | 8 h 39 min |

The resulting DEM (Figure 6) provided a high-resolution data set which was used to characterize the terrain morphology and with a GIS software, QGIS 3.12.3, we created a slope map, Figure 7, which allowed us to identify and classify the terrain according to its slants, and this was the basis to define the circulations and the exterior areas in combinations with analogue cases.

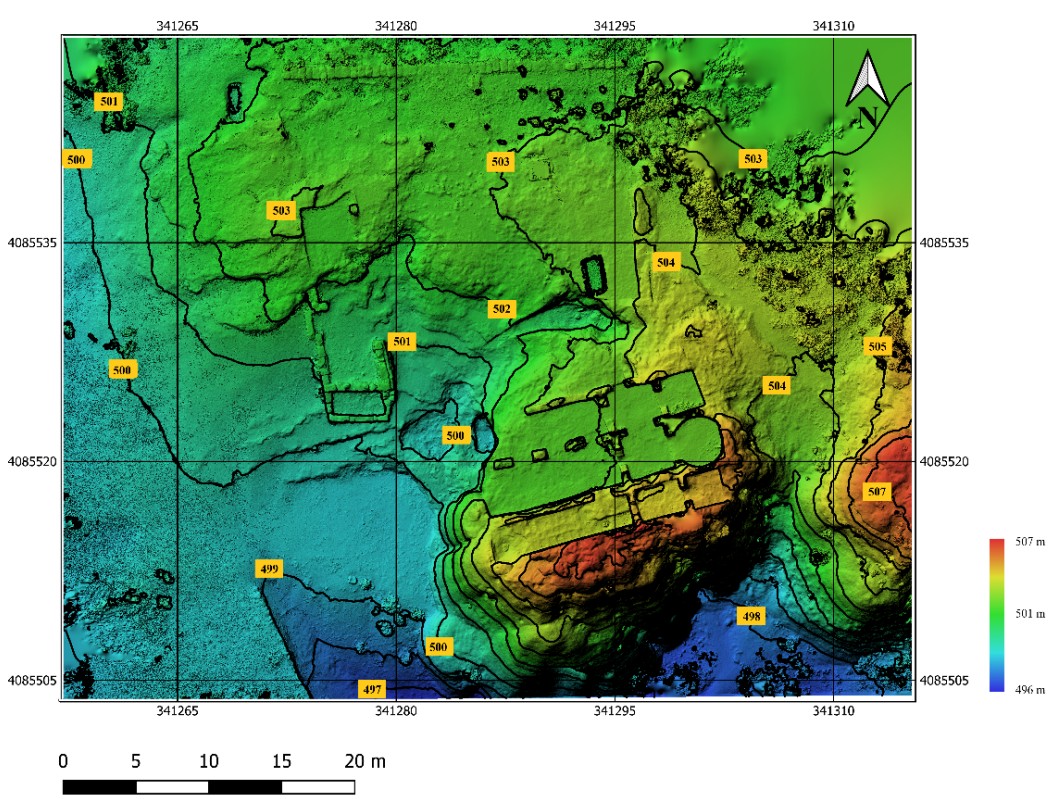

**Figure 6.** Bobastro Digital Elevation Model. The coordinates (m) are UTM, zone 30, referred to ETRS89.

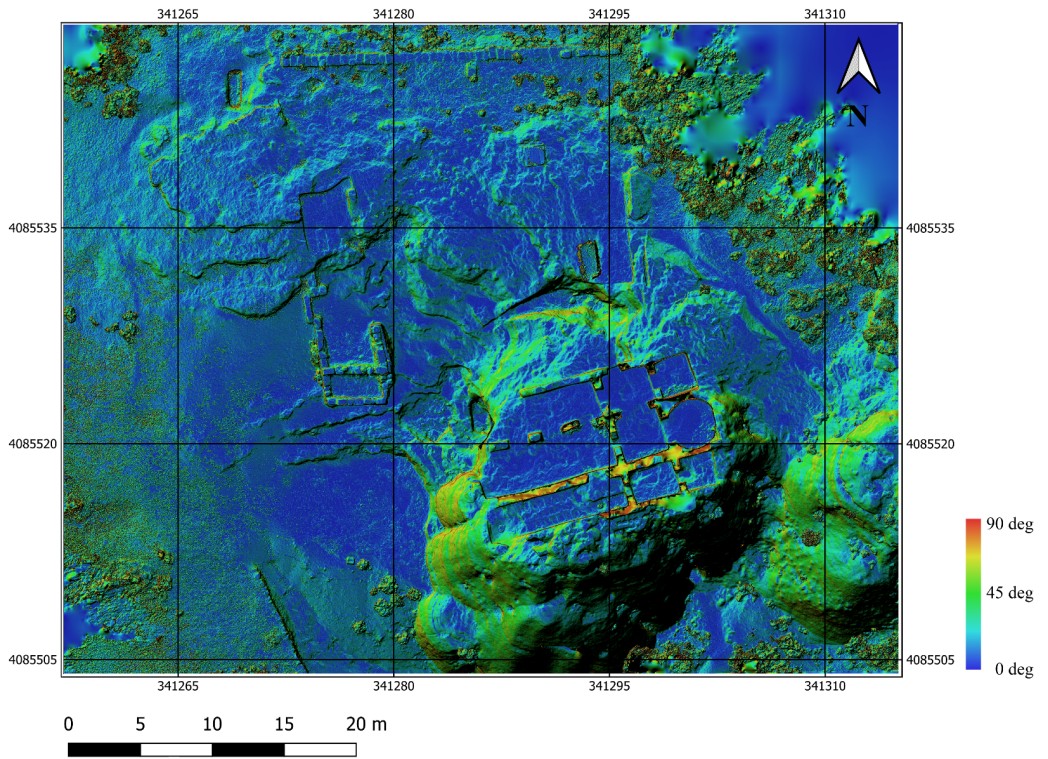

**Figure 7.** Bobastro Digital Slope map. The coordinates (m) are UTM, zone 30, referred to ETRS89.

In addition, the orthomosaic helped us to fix the main grids and site references. The remains of buildings and construction references in the past were used as the main axes; defining the walls carved in the longer stones, more complete sections and those that form the perimeter of the building. These axes helped to define the envelopes and the main areas of the site.

Starting from there, lines were drawn according to the vestiges found on the site, delimiting the temple, as well as the cloister according to the reference of the massive walls in the place. Having these references, the unevenness between the two areas was noted, together with the orientation of the slope, which was reinterpreted as follows: the minimum slope is delimited with the water mirror at the center of the cloister, which simultaneously has interconnecting walks. The area with the steep slope presents different reliefs which show signs of stairs, thus modeling the connectivity between areas on different high levels.

On the Northern side, we were able to identify a wall, which corresponds with the cloister area limits, the circular entrance of a cistern, and the main elements of the basilica that are marked in Figure 8.

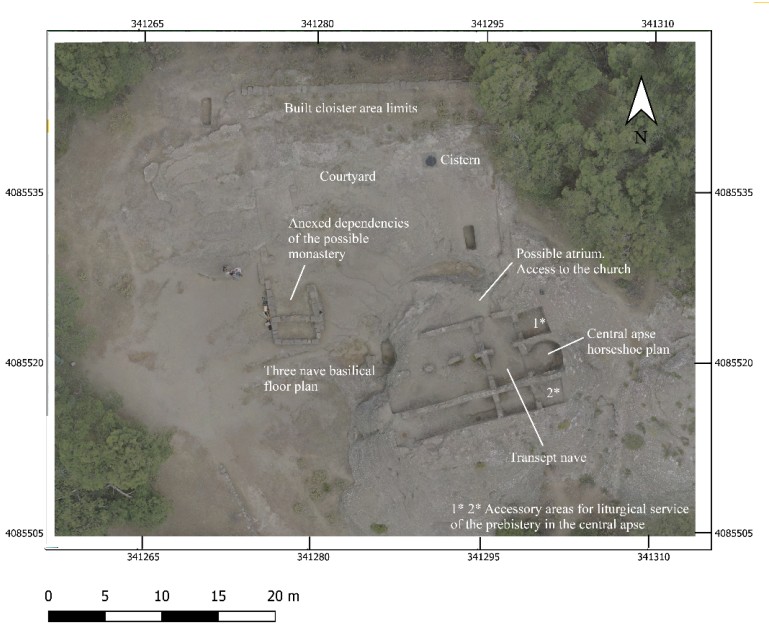

**Figure 8.** Orthomosaic of Bobastro archaeological site. The coordinates (m) are UTM, zone 30, referred to ETRS89.

In addition, we established a linear circulation that starts from the bottom of the land until reaching a half level, where the flight of stairs parallel to the perimeter of the tower communicates with the temple and its upper areas. From the middle level, a communication is established to a rest area that delimits a fork in the paths, the path being oriented to the east, which allows entry to the temple and the cloister at the end of the tour. On the other hand, the path facing north allows a communication towards the water mirror, which has three platforms according to the topography of the place. The final result can be seen in Figure 9.

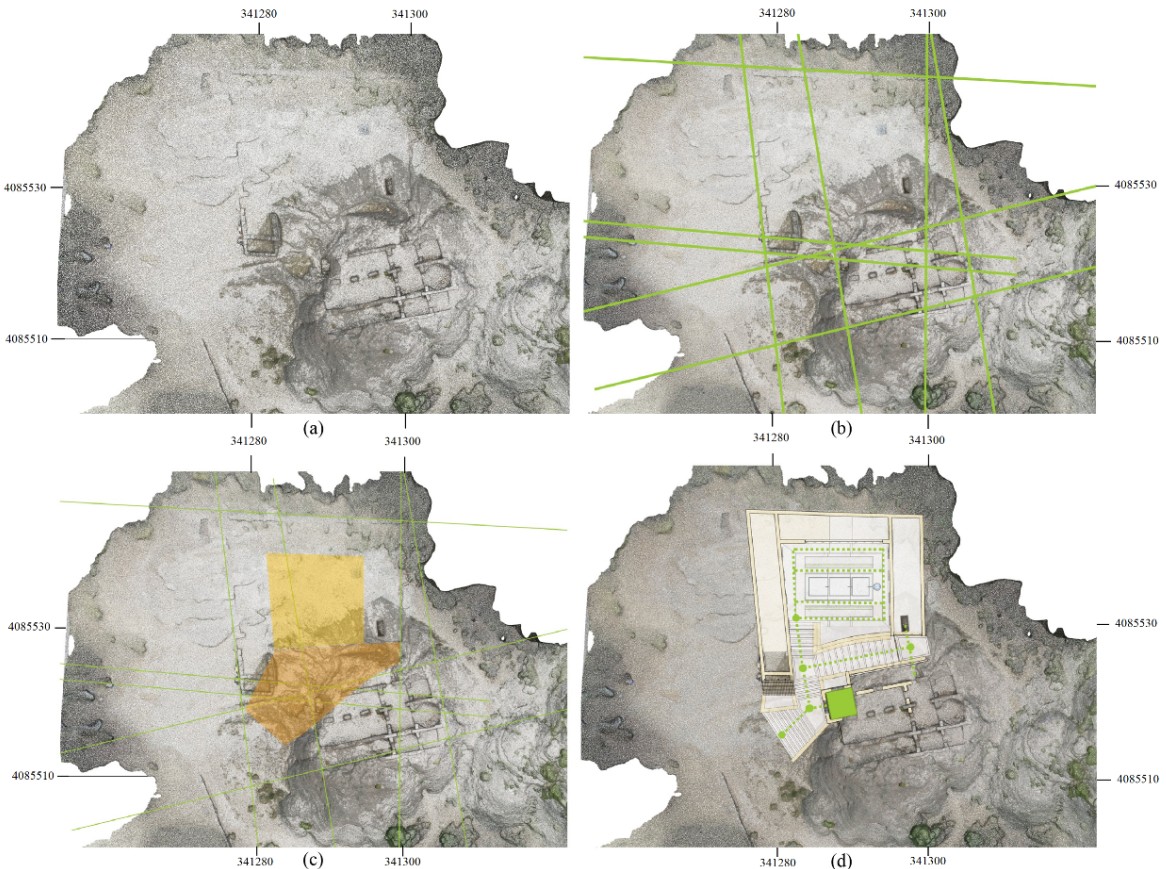

**Figure 9.** Clockwise from top left to right: (**a**) the orthomosaic, (**b**) main axis and site references, (**c**) the two slopes zones: Low (gold) and moderate (brown) and (**d**) circulation. The coordinates (m) are UTM, zone 30, referred to ETRS89.

## 5. Discussion

Without discarding the contributions of traditional archaeology [10,12] a century later, we used a new methodology that facilitates the analysis and proposal of interpretive hypotheses that help to restore the lost images of what during almost a century have been abandoned, robbed, and abused ruins, in order to increase its value.

The results obtained from the contour map and the generated orthophoto corroborate the interpretations made of a religious complex with a main space in basilical typology semi-excavated in the rock as seen in Figure 10. This type of floor plan is no stranger to Mozarabic religious architecture since the cases of basilical schemes of three naves with a differentiated chevet, with horseshoe or straight apses, are comparable. The subtle changes in the level isolines detected in that space are perfectly compatible with the practice of a solid foundation to raise the building by emptying the bedrock. This characteristic is comparable to temples that share a similar chronology and even greater geographical proximity [24].

The analysis of the resulting data shows a different hypothesis regarding the area built as a cloister attached to the main building. There are interior spaces that are arranged around an open space, which must have been residential and multipurpose, connected with a small monastery. This reaffirms again the correspondence with similar eremitical practices that had a long tradition of rock exploitation. An example of these places is the Mesas de Villaverde that presents suitable geomorphological characteristics for it [25].

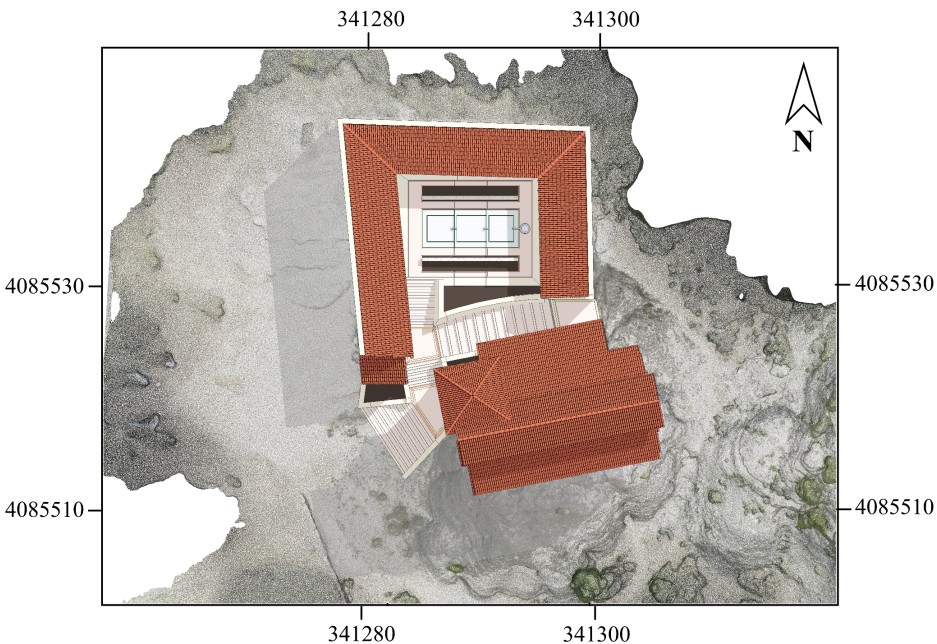

**Figure 10.** Proposed final aspect. The coordinates (m) are UTM, zone 30, referred to ETRS89.

However, there are some additional elements that must be considered in parallel to the interpretation of data obtained in order to propose the most plausible reconstruction hypothesis. The work was carried out on the rock delimiting a perfectly proportioned ecclesiastical space in which complex elements have been designed at the planimetric level: transept, central apse with a correct horseshoe layout, cruciform pillars, and delimitation of accessory spaces for ancestor worship, interpreted as a prosthesis and a diaconicon (it must be considered that the model of monastery that would be imposed later in the Romanesque period had not yet become general).

In the orthophoto, in the north, the remains of a built lateral structure were identified. That could function as an atrium and in which an important empty burial was located. This interpretation is acceptable from the funeral perspective of this period, in which the prohibition to bury in sacred soil prevailed. In this sense, important personalities tried to get as close as possible to their burial places, although without crossing them [26], such as the case of Umar Ibn Hafsun and his son, of which the Andalusian chronicles recorded their exhumation as punishment. From this nave-atrium, a circulation system could be established around an open courtyard in which the presence of water had to be ensured, since the remains of a cistern were also found.

The perimetrical walls of the cloister spacing that can be seen both on the isolines map and in the orthophoto do not reflect the sufficient potency or the existence of pressure-offset structures. This points to a light covering based on simple timber roof frames, possibly splayed coping that would be protected by ceramic tiles, the remains of which appeared in previous archaeological campaigns in full accordance with the building practice of Andalusian constructions. The horseshoe arches that are defined on the plan and the elevation of the basilica also correspond to these characteristics, due to the bulge that its development shows, close to the emirate horseshoe type.

The similarities with the Cordovan artistic style are even more evident in constructive and decorative details as the almagra sand paving appeared latter in 2001 in another church in Bobastro, closer to the palatine residential area [27] and the remains (glazed ware, fragments of an epitaph, and a capital cataloged as Bobastro fortress) currently preserved in different museums in Malaga, the National Archaeological Museum, or the archaeological site of Granada. Despite the dating difficulties of these pieces, their presence and variety leads us to rethink that the cloister gallery had more open fronts with gateways with columns of a certain artistic quality [28], similar to that obtained by Cordovan ateliers. The works on traditional materials such as stone, brick, or tile and wood [25]

were techniques extended throughout the extensive Andalusian territory, regardless of whether it was attached to a Christian house of worship building or not.

The denominations that have been given to this Bobastro complex have been varied: fortified city, capital of the Hafsunis and even episcopal see [27]. Its short life and subsequent destruction and reconstruction after the conquest by Abd al-Raḥmān III have made it the subject of various investigations that have recently focused on other areas, forgetting the ruins of the convent. Despite the hypothetical restitution of its plant, detailed studies and even a volumetric restitution of its general appearance, in an explanatory poster next to the ruins, a definitive interpretation of the remains has not given yet [29].

## 6. Conclusions

Geomatics offers a wide variety of new tools that are changing the way of working in many disciplines, archaeology being one of them. The fusion of technologies such as sensors integrated in UASs, photogrammetry, GNSS, and GIS tools has been used in this paper in order to obtain a 3D reconstruction of archaeological remains. In this sense, 3D modeling is improving accuracy and efficiency in recording methods. In fact, these models allow the archaeologist to take measurements and carry out analysis of the objects properties, their location, and landscapes around them with a level of detail and accuracy as never seen before. The combination of techniques in the 3D survey taken at Bobastro can help the archaeologist not only to register all details of archaeological remains but also to identify structures that are hidden from our view. The obtained results show a more detailed description of the Bobastro Mozarab Church that could have been constructed. The presence of a cistern in the cloister, the different high levels of heights, and the smooth slope between them suggest the existence of some kind of fountain and stairs communicating with both spaces. Gathering all this evidence, we can confirm the presence of a small monastery around the basilica, which suggests that the complex was even more complex and richer than it was believed. In the field of archaeology, the collecting data process must be very meticulous and formally registered. An enormous amount of information can be obtained on an archaeological site in just one day. As insignificant as a piece of data or a piece of pottery may seem, it can shed light on the research carried out and therefore must be recorded, otherwise it will be lost forever. An adequate record of archaeological remains is not only necessary to be studied and thus also protected, but they are also a source of income for the conservation of the cultural heritage of society. Knowledge of the past helps to understand the present and thus build a better future. This is why it is necessary that the historical heritage be studied and properly preserved so that, generation after generation, it is transmitted without its essence being blurred. Thus, it is necessary to find a balance between the loss of the material remains against their significance, and the ability of the practicing archaeologist to produce a measured, drawn, and written record; appropriately conserved and archived finds; and a fully synthesized final report lodged with the relevant authorities [30]. Finally, the increase and availability of the Internet around the world opens innovative ways of cooperation. For instance, the cloud allows for storing all results and makes them available to researchers and students all over the world. Even more, as results are 3D metric data, they can be used as a basis for Virtual Reality and Augmented Reality applications in Cultural Heritage.

**Author Contributions:** Data curation, C.E.; Formal analysis, M.J.C., G.E., P.E., and M.I.R.; Funding acquisition, D.R. and F.R.F.; Investigation, C.E. and F.R.F.; Methodology, C.E., G.E., and M.P.; Software, C.E., A.B., G.E., P.M., E.O. (Erick Oliva), E.O. (Edgar Osla), and S.S.; Supervision, D.C., F.R.F.; Visualization, G.E.; Writing—original draft, C.E., J.M.J., M.J.C., G.E., and M.I.R.; Writing, review and editing , C.E., J.M.J., M.J.C., G.E., M.I.R., and F.R.F. All authors have read and agreed to the published version of the manuscript.

**Funding:** This research has been partially financed by the Ministerio de Economía y Competitividad and the European Union (via ERDF funds) through the research project TIN2017-84968-R.

**Acknowledgments:** The authors wish to express their thanks to Javier Rey and José Lisazoain for their help and their valuable comments.

**Conflicts of Interest:** The authors declare no conflict of interest.

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
