# Peer review of "The UAS-Based 3D Image Characterization of Mozarabic Church Ruins in Bobastro (Malaga), Spain"

_remotesensing, doi:10.3390/rs12152377_

Round 1

Reviewer 1 Report

Dear authors,

The manuscript is interesting about new uses in UAV to get 3D models in churches as you showed. However, I have an unique point to discuss with you. What was the reason to choose only 2 check points to evaluate the accuracy of your 3D Model?.

The table 2 is showing control points, however when you evaluate the accuracy in a model, you have only to include the check points (ASPRS Accuracy Standards for Digital Geospatial Data, 2015) and I do not if 2 check points are sufficient to evaluate all the accuracy in the model. Maybe, you have to include more points near to your actual control points or in a good distribution.

Thank you.

Reviewer 2 Report

The article’s theme is very interesting but the text presents several critical points. The aerial image of the area in Fig. 1 doesn’t refer to its origin, what is the source? The image processing is adequately explained but in the discussion it would be appropriate to insert detailed images of the various steps indicated in the text. There aren’t suitable bibliographical references to support the discussion. It’s advisable that the authors have the article proofread by a professional English translator or by a native speaker. The revisions are required before publishing.

Reviewer 3 Report

Though not native English, I would suggest a thourough redaction and correction of the paper. Phrases are sometimes very long and not easy to follow, for the reader; there are mistakes (single-pluriel; Archeology doesn't need a capital, Fianlly... finally -double use, etc.); evidences 'the slopes map ....allows us to identify the different slopes on the site... douh!).

The paper presents a project where UAV technology has been used on the ruins of a mozarabic church in Malaga; this was realized over a rather small but relevant area (2500m²). The purpose of the study is the reconstruction and visualization of the site, to define and identify the whole structure of the site (a church and a cloister).

The introduction is clear and refers to state of the art papers, though of course one could mention ten other papers also...

The technology used is basic, with SfM and QGis, marking GCPs, making a slope map and an orthomosaic. So basically this is not orginal. Question is whether this 'not original' study applied to a monument of this type can be considered as a novelty. Thus is the quite long descritpion of the way to get data not really a necessitiy, as it doesn't bring any new information. It would be better to give the basic information (flying time, amount of photos, cameras etc.) in a short version, and maybe focus on aspects that are not 'the normal'.

I think the section 3. Methods needs redaction and esp. reorganisation.

Concerning the results, it seems to me that the outcomes are OK (DEM, Slope, Orthomosaic). Of course, in the discussion of these results, it appears that having a new set of (visual) data helps the interpretation of the site, but I think one should make more clear what the difference is between the results of one century ago and those of today. It is not clear to me that a lot of major new elements appeared. Make this more clear (if so), or focus more on the possibility for a better site heritage management, maybe.

The conclusions draw some basic lines (archaeologyical digging is a destructive process, images and new data can make access ot the site (sensibilisation and local development), the cloud allows to share...). Is this necessary for this paper? Focus on the conclusions that are those of the research realised here.

Round 2

Reviewer 1 Report

Thank you for your answer

Reviewer 2 Report

The new version has a better description of the contents. The improved images and tables quality is very appreciable, especially the final reconstruction proposal is very interesting (Fig. 10) and gives a convincing view of the original structure. Finally, it was important to implement the bibliography, for greater scientific soundness.

Reviewer 3 Report

The authors did a quite thorough rewriting of their paper, taking into account the remarks I made in the first review. 

The redaction of the English text improved seriously. 

So I do not have any problem anymore to see this paper published.